# The Uprise of Human Leishmaniasis in Tuscany, Central Italy: Clinical and Epidemiological Data from a Multicenter Study

**DOI:** 10.3390/microorganisms12101963

**Published:** 2024-09-27

**Authors:** Anna Barbiero, Michele Spinicci, Andrea Aiello, Martina Maruotto, Roberta Maria Antonello, Giuseppe Formica, Matteo Piccica, Patrizia Isola, Eva Maria Parisio, Maria Nardone, Silvia Valentini, Valentina Mangano, Tamara Brunelli, Loria Bianchi, Filippo Bartalesi, Cecilia Costa, Margherita Sambo, Mario Tumbarello, Spartaco Sani, Silvia Fabiani, Barbara Rossetti, Cesira Nencioni, Alessandro Lanari, Donatella Aquilini, Giulia Montorzi, Elisabetta Venturini, Luisa Galli, Giada Rinninella, Marco Falcone, Federica Ceriegi, Francesco Amadori, Antonella Vincenti, Pierluigi Blanc, Iacopo Vellere, Danilo Tacconi, Sauro Luchi, Sara Moneta, Daniela Massi, Michela Brogi, Fabio Voller, Fabrizio Gemmi, Gian Maria Rossolini, Maria Grazia Cusi, Fabrizio Bruschi, Alessandro Bartoloni, Lorenzo Zammarchi

**Affiliations:** 1Department of Experimental and Clinical Medicine, University of Florence, 50121 Florence, Italy; anna.barbiero@unifi.it (A.B.);; 2Infectious and Tropical Diseases Unit, Careggi University Hospital, 50134 Florence, Italy; 3School of Human Health Sciences, University of Florence, 50121 Florence, Italy; 4Infectious Diseases Unit, Department of Multidimensional Medicine, Santa Maria Annunziata Hospital, Bagno a Ripoli, 50012 Florence, Italy; 5Laboratory Medicine Unit, Azienda USL Toscana Nord Ovest, 56121 Livorno, Italy; 6Clinical Pathology and Microbiology Unit, San Luca Hospital, 55100 Lucca, Italy; 7Laboratory of Chemical-Clinical Analysis, San Luca Hospital, 56121 Lucca, Italy; 8Infectious Disease Department, Misericordia Hospital, 58100 Grosseto, Italy; 9Translational Research and New Technologies in Medicine and Surgery, University of Pisa, 56126 Pisa, Italy; 10Microbiology Unit, S. Stefano Hospital, 59100 Prato, Italy; 11Microbiology Unit, S. Jacopo Hospital, 51100 Pistoia, Italy; 12Department of Medical Biotechnologies, University of Siena, 53100 Siena, Italy; 13Infectious and Tropical Diseases Unit, Azienda Ospedaliero Universitaria Senese, 53100 Siena, Italy; 14Infectious Diseases Unit, Ospedali Riuniti di Livorno, 57124 Livorno, Italy; 15Infectious Diseases Unit, Ospedale Misericordia, 5810 Grosseto, Italy; 16Infectious Diseases Unit, S. Stefano Hospital, 59100 Prato, Italy; 17Infectious Diseases Unit, Meyer Children’s Hospital IRCCS, 50139 Florence, Italy; 18Section of Pediatrics, Department of Health Science, University of Florence, 50121 Florence, Italy; 19Infectious Diseases Unit, Department of Clinical and Experimental Medicine, Pisa University Hospital, University of Pisa, 56126 Pisa, Italy; 20Infectious Diseases Unit, Nuovo Ospedale Apuane, 54100 Massa, Italy; 21Infectious Diseases Unit, S. Jacopo Hospital, 51100 Pistoia, Italy; 22Infectious Diseases Unit, Ospedale San Donato, 52100 Arezzo, Italy; 23Infectious Diseases and Epatology Unit, San Luca Hospital, 55100 Lucca, Italy; 24Section of Pathology, Department of Health Sciences, University of Florence, 50121 Florence, Italy; 25Unit of Infectious Diseases, Department of Medical Specialties, Empoli, 50129 Florence, Italy; 26Unit of Epidemiology, Regional Health Agency of Tuscany, 50141 Florence, Italy; 27Quality and Equity Unit, Regional Health Agency of Tuscany, 50141 Florence, Italy; 28Microbiology and Virology Unit, Careggi University Hospital, 50134 Florence, Italy; 29Programma di Monitoraggio delle Parassitosi e Formulazione di Nuovi Algoritmi Diagnostici, AOU Pisana, 56126 Pisa, Italy

**Keywords:** leishmania, Italy, epidemiology, phlebo-borne disease, climate change, One Health

## Abstract

Human leishmaniasis is facing important epidemiological changes in Southern Europe, driven by increased urbanization, climate changes, emerging of new animal reservoirs, shifts in human behavior and a growing population of immunocompromised and elderly individuals. In this evolving epidemiological landscape, we analyzed the clinical and epidemiological characteristics of human leishmaniasis in the Tuscany region of Central Italy. Through a multicentric retrospective analysis, we collected clinical and demographic data about all cases of leishmaniasis recorded between 2018 and 2023. We observed 176 cases of human leishmaniasis, with 128 (72.7%) visceral leishmaniasis (VL) and 47 (26.7%) cutaneous leishmaniasis (CL). Among these, 92.2% of VL and 85.1% of CL cases were autochthonous. The cumulative incidence of autochthonous human leishmaniasis was 0.22 cases per 100,000 inhabitants in 2018, but reached 1.81/100,000 in 2023. We identified three main areas of transmission: around the city of Florence (North-East Tuscany), around Grosseto city (South-West Tuscany) and Elba Island. Our findings confirm that the epidemiology of leishmaniasis is undergoing significant changes in Central Italy. Awareness towards this emerging health threat and surveillance strategies need to be improved in order to reliably assess the disease’s burden. Further research is needed in a “One-Health” perspective, to clarify the epidemiological dynamics at the environmental, reservoir, vector and human levels. The role of climate change and specific climatic factors affecting the epidemiological patterns of human leishmaniasis should be assessed. Further knowledge in these fields would promote targeted control and prevention strategies at regional and national levels.

## 1. Introduction

Human leishmaniasis is caused by flagellate protozoans of the genus *Leishmania*, transmitted to humans through the bite of phlebotomine sandflies. Leishmaniasis is likely highly under-reported worldwide due to a lack of sources and effective national surveillance plans, especially in low-middle-income-countries, as well as poor awareness among clinicians about the importance of reporting infectious diseases subject to national or international surveillance. Despite underestimations, approximately one million new cases of human leishmaniasis are reported to occur worldwide each year, predominantly in tropical and subtropical regions [1]. The infection can affect humans in two main clinical forms, tegumentary leishmaniasis and visceral leishmaniasis (VL) [2], the latter being associated with high hospitalization and fatality rates, which can reach 95% if untreated [1]. The mortality rate of VL is increased by factors such as availability of appropriate treatments (especially in low-resource settings), late diagnosis, malnutrition and HIV co-infection [1].

Tegumentary leishmaniasis includes cutaneous (CL), the less common mucocutaneous (MCL, usually observed in the New World) and mucosal leishmaniasis (ML, which can be observed in the Old World). TL is not associated with fatal outcomes (fatality rate < 1%), but may lead to disfiguring and dysfunctional skin lesions, causing significant stigma and adversely affecting quality of life [3].

In Southern Europe, human leishmaniasis is facing important epidemiological changes, driven by factors such as increased urbanization, climate changes, migration phenomena and a growing population of immunocompromised and elderly individuals. These elements do not only favor the disease spread in specific areas, but also hinder the implementation of effective control strategies [2].

In Italy, the only endemic *Leishmania* species is *Leishmania infantum,* which is responsible for both tegumentary and visceral forms. In the Italian peninsula, leishmaniasis is endemic in Southern regions, the major islands (especially Sicily), the Tyrrhenian coast (Tuscany and Liguria regions) and, to a lesser extent, the Adriatic coast (Emilia-Romagna Region) [4]. Historically, the main leishmaniasis reservoir has been the dog, which can develop severe and often fatal forms of the disease [5].

Since the 1990s, a northward spread of both CL and VL has been documented in Italy, with epidemic foci particularly affecting the North-Eastern regions and, in particular, the Emilia Romagna region [6,7,8]. Recent studies conducted in that region have also highlighted the potential role of sylvatic mammals as additional animal reservoirs in the zoonotic transmission cycle of human leishmaniasis. Rugna et al. proved that different *Leishmania* strains circulate in human and dogs in the Emilia-Romagna Region [9], whereas Magri et al. reported that over 90% of *L. infantum* strains circulating in roe deer were overlapping with those affecting humans in the same area [10].

In the Tuscany region (Central Italy), data reported by the Health Regional Agency, based either on hospitalization records or mandatory notification reports, indicate a significant and progressive increase in the number of human cases. Annual VL cases ranged from 2 to 7 until 2020, but increased to 14 in 2021 and 26 in 2022. The disease burden is likely underestimated when considering only notification reports data, as 35 VL hospitalizations were reported in 2022, in contrast with the only 26 notification reports in the same year. Similarly, the number of CL cases has doubled over the past decade regionwide [11].

In this shifting epidemiological scenario, we conducted a multicenter retrospective study to describe the clinical and epidemiological characteristics of human leishmaniasis cases recorded in the Tuscany region between 2018 and 2023. Additionally, we aimed to identify areas with higher transmission rates, with the objective of collecting data that could inform the future implementation of effective surveillance and control strategies in this region.

## 2. Materials and Methods

This study was promoted by the Tuscany Reference Centre for Tropical Diseases (TRCTD) and included all Infectious Diseases units (*n* = 12) of the Tuscany region.

We conducted a retrospective multicenter study, including all patients diagnosed with CL, VL or ML, according to IDSA/ASTMH guidelines [12], observed in one of the twelve Infectious and Tropical disease units between 2018 and 2023. By consulting digital records of both inpatients and outpatients, we collected clinical and demographic data of enrolled subjects.

To reduce underestimation of the actual number of cases during the study period and to identify cases managed outside Infectious Disease units, we also collected leishmaniasis diagnoses made in microbiology, parasitology and pathology laboratories, hospital discharge forms reporting diagnoses of leishmaniasis and surveillance reports of leishmaniasis cases recorded in each center.

Patients of all ages, including pediatric patients, were included if they had a parasitologically confirmed diagnosis of CL, VL or ML. Clinical, epidemiological and demographic information of the observed cases was collected using REDCap 8.11.6. (Project REDCap, USA); the platform was properly configured for multicenter data processing and data validation was conducted.

We established presumptive sites of infection based on suggestive clinical history (e.g., symptoms onset following a stay in a specific vacation area). When no suggestive epidemiological elements were observed in the clinical history, we considered the domicile (i.e., the place where the patient spent most of his time) as the most presumptive site of infection. We defined autochthonous episodes as those where the presumptive site of infection was within the study area (Tuscany region), whereas imported cases were those where the infection was most likely acquired in other Italian regions or outside Italy. As regarding complications, hemophagocytic lymphohistiocytosis (HLH) was confirmed when diagnostic criteria were met [13,14].

### 2.1. Statistical Analysis

A descriptive analysis was conducted to analyze clinical and epidemiological characteristics of the enrolled population.

Continuous variables were described as medians and inter-quartile-range (IQR), whereas categorical variables were evaluated as frequencies and proportions. Association between categorical variables was evaluated with Chi-square test or Fisher’s exact test where appropriate, and continuous variables with Wilcoxon rank-sum test. Results were considered statistically significant for *p*-values < 0.05. The software STATA v18.0 (STATACorp, USA) was used for descriptive statistical analyses. The absolute number of cases observed each year was calculated, and the annual incidence (no. cases/100,000 inhabitants) of CL and VL in the study area was analyzed, taking into account the annual regional population, as reported by the National Institute of Statistics [15].

Relationships between age, gender, immunosuppression, exposure risk factors (i.e., owning dogs and practicing frequent outdoor activities) and type of leishmaniasis were analyzed through a logistic regression model in order to reduce confounding variables.

Geographical coordinates were assigned to the presumptive sites of infection for each recorded case, and Kernel density estimation was applied to analyze the geospatial distribution of autochthonous cases in the study area. Kernel density estimation and geographical distribution of presumptive sites of infection were represented with a heatmap, using the freely available server Heatmapper [16]. For cases where no suggestive site of infection was identified according to anamnestic information, the domicile was considered the presumptive site of infection.

### 2.2. Ethical Considerations

Enrolled patients gave informed consent for participation to the study, and each record was pseudo-anonymized before compilation of the Case Report Form (CRF). The study was performed in accordance with the ethical principles of the Declaration of Helsinki and with the International Conference on Harmonization Good Clinical Practice guidelines. Data collection was approved by the local ethics committee “Comitato Etico Regione Toscana—Area Vasta Centro” (protocol code: TOSMANIA_2023; approved on 27 February 2024; registration code: 25425_oss).

## 3. Results

Overall, 202 cases of human leishmaniasis were observed during the study period. Among these, 26 were relapsing episodes, while 176 were first episodes. Of the 176 first episodes, 128 (72.7%) were VL, 47 (26.7%) were CL and 1 case (0.6%) was ML.

According to clinical history, 118/128 (92.2%) cases of VL were autochthonous, while 9/128 (7.0%) were imported from other countries, including Albania (2), Cuba (1), Kenya (1), Oman (1) and Romania (1) (it was not possible to determine infection country in three cases due to travel to several countries before infection). One case (0.8%) was imported from another Italian region (Campania).

Regarding CL, we observed 40/47 (85.1%) autochthonous cases, 5/47 (10.6%) imported cases from other countries (Mexico (2), Nicaragua (1), Peru (1), India (1)) and 2 cases (4.3%) imported from another Italian region (both from Sicily).

No significant difference in terms of proportion of imported cases was observed when comparing the CL and VL cohort (*p* = 0.161).

The only ML observed during the study period was autochthonous. Given that only one ML case was reported, further analysis on epidemiological and clinical characteristics focused on VL and CL cases.

As reported in Figure 1, the number of VL cases increased over the study period, from 7 cases recorded in 2018 (incidence: 0.19/100,000) to 43 cases reported in 2023 (incidence: 1.18/100,000). Similarly, 1 CL case was observed in 2018 (incidence: 0.03/100,000), reaching 23 cases in 2023 (incidence: 0.63/100,000). The cumulative incidence of human leishmaniasis (including both CL and VL) increased from 0.22/100,000 in 2018 to 1.81/100,000 in 2023.

Annual absolute numbers of VL and CL imported cases are also shown in Figure 1.

Based on the available data from clinical history, geographical coordinates of presumptive sites of infection were obtained for 99/118 autochthonous VL cases and 27/40 autochthonous CL cases. The distribution of these presumptive sites of infection and Kernel density estimation are represented in Figure 2. Three main areas were detected as having higher transmission rates: (1) the area around the city of Florence and especially in its South-Eastern side (Figure 2—X), (2) the area surrounding the city of Grosseto (Figure 2—Y), located in South-Western Tuscany region and, to a lesser extent, (3) Elba Island (Figure 2—Z).

Among the 158 autochthonous cases of VL and CL, 114 (72.2%) were males. A value of 33 out of 158 (20.9%) subjects reported owning at least one dog; of these, 2 out of 33 reported a history of leishmaniasis in their dog. The median age was 58.7 (IQR: 25.3–73) years, with the VL population being significantly older (*p* = 0.015) than the CL population. Demographic characteristics and risk factors of both VL and CL populations are reported in Table 1.

Case distribution by age group for both CL and VL leishmaniasis is shown in Figure 3. While the highest number of cases was observed in the age groups 1–10, 60–70 and 70–80 for VL, the age groups 0–10, 30–40 and 60–70 were the most affected by CL.

The median time to diagnosis was 21 days (IQR: 12–37) for VL and 174 days (IQR: 99–262) for CL. The time to diagnosis for VL was significantly longer (*p* = 0.0174) in the immunocompromised population (28.5 [IQR: 16–58.5] days) compared to the non-immunocompromised population (18.5 [IQR: 11.5–27.5] days). No significant difference (*p* = 0.930) in median time to diagnosis was observed for CL when comparing the immunocompromised and non-immunocompromised population.

Regarding hospital admission, 109/118 (92.4%) subjects with VL and 16/40 (40.0%) with CL were hospitalized. The median duration of hospitalization was 13 days (IQR: 8–21) for VL and 2 days (IQR: 2–62) for CL.

Data on the most common complications associated with VL showed that 32/118 (27.1%) subjects had HLH, 23/118 (19.5%) needed blood component transfusion and 11/118 (9.3%) had bacterial superinfections. Correlation between observed complications and VL was established according to temporal correlation between VL and complication’s onset, patient clinical history and exclusion of other possible causes for such complications.

The enrolled subjects had a median follow-up time of 6 months (IQR: 6–12) for VL and 12 months (IQR: 6–12) for CL. Outcomes at 3, 6 and 12 months after treatment for both VL or CL are reported, based on available information, in Table 2. As reported in the table, only 52/94 and 11/28 patients affected by VL and CL, respectively, completed a 12-month follow-up.

Regarding VL cases, 9 deaths were observed among the 70 patients with complete follow-up at 6 months (case fatality rate 12.9%). Reason of death was reported for 7/9 patients, with 4/7 deaths due to leishmaniasis and 3/7 deaths due to other reasons.

The median age of the nine deceased patients was of 72.0 [IQR: 64.3–86.9] years, being significantly older than the general VL population (*p* < 0.005). Three of them (37.5%) were immunocompromised, one of which with immunodepression due to HIV infection. As regarding associated complications, 3/8 had need of hemoderivative transfusion, 5/8 had HLH and 1/8 had bacterial superinfection.

Mandatory surveillance reports were filled out for 115/158 (72.8%) autochthonous cases (75.4% for VL and 65.0% for CL).

## 4. Discussion

This study describes and analyzes the clinical and epidemiological features of human cases of leishmaniasis observed in the Tuscany region between 2018 and 2023. Approximately 2.6 million inhabitants live in this region, and around 14.6 million tourists visit Tuscany each year [17,18].

Although Italy has historically been endemic for leishmaniasis, several epidemiological aspects of the disease have been changing in recent years. Despite recent reports indicating an overall decrease in the number of leishmaniasis cases observed at the country level [19], national-level data may mask regional differences related to specific epidemiological and environmental factors. Indeed, the disease has been increasingly reported in previously non-/low-endemic areas, such as the Central and North-Eastern regions [6,7,8,20,21,22,23,24,25,26,27,28,29].

In this shifting epidemiological context, our research confirmed a significant increase in incidence of autochthonous cases of both VL and CL in the study area. The overall incidence of human leishmaniasis rose from 0.22 cases per 100,000 inhabitants in 2018 to 1.81/100,000 in 2023. This rate is consistently higher than the 0.74 incidence recently reported by Maia and collaborators for the Tuscany region based on mandatory surveillance reports and hospitalization reports recorded between 2011 and 2016, confirming the emergence of this health issue in recent years [19].

Notably, despite the limitations associated with a low number of cases and the potential bias due to the travel restriction during the COVID-19 pandemic, no significant increase in imported cases of either CL or VL was observed over the study period. Along with the reported overall decrease in human leishmaniasis cases in Mediterranean countries in the past years [19], these data suggest that the uprise of the disease could be confined to specific geographical areas and related to local epidemiological and environmental factors that need further exploration. Indeed, the rise in human leishmaniasis cases in the Tuscany region could be attributed to several factors.

First, climate changes could have impacted the epidemiological trend. Evidence suggests that climate changes can expand areas suitable for sandflies, both in latitude and altitude, thereby affecting the epidemiology of human leishmaniasis [22,23]. Increasing urbanization, migration and the global movement of dogs and livestock are also important determinants of the disease’s epidemiological changes [24]. Changes in human habits can also have a relevant impact. For example, during the COVID-19 pandemic, arthropod-borne diseases such as tick-borne encephalitis increased in Germany, likely due to more frequent outdoor activities resulting from restrictions imposed on enclosed places [25].

Moreover, increasing evidence indicates the involvement of mammals other than dogs, such as roe deer, in the life cycle of the parasite in the same North-Eastern Italian regions where recent outbreaks have been reported [9,10]. Similarly, data from another Southern European country, Spain, indicate the role of urbanization of rural areas and of sylvatic leishmaniasis reservoirs (hare) in the Fuenlabrada outbreak [26,27]. This suggests that land use modification (e.g., deforestation, extension of intensive agriculture lands) and sylvatic vertebrates could contribute to the epidemiological changes occurring in some Mediterranean areas, highlighting the need for further research in this field.

Further research is needed that analyzes climatic and environmental anthropogenic factors, along with epidemiological characteristics of the infection at both animal and vector levels, and their possible connection to the described epidemiological changes, in order to better understand this phenomenon from a “One-Health” perspective.

Our report showed a higher prevalence of VL compared to CL, with VL representing 74.7% of all autochthonous cases. No significant differences in the proportion of imported cases were observed between the two groups. This finding aligns with existing literature, which reports 82% of VL cases and 14% of CL cases observed in Italy between 2005 and 2020 [19].

The geospatial distribution of autochthonous cases was analyzed by collecting the geographical coordinates of likely sites of infection and mapping their distribution in the studied area. Using Kernel density estimation analysis, three main areas with higher transmission rates were identified: (1) the area between the cities of Florence and Bagno a Ripoli, located in the South-Eastern side of Florence (Figure 2—X), (2) the area surrounding the city of Grosseto (Figure 2—Y), located in South-Western Tuscany and, to a lesser extent, (3) Elba Island (Figure 2—Z). The high concentration of cases around the city of Florence could partly be biased by the high population density in this area; however, the concentration of cases in other areas of the Tuscany region, where there is not a relevant population density, does suggest that leishmaniasis has a specific distribution pattern in the study area.

Regarding the demographic characteristics and analyzed risk factors for the autochthonous cases, 74.6% of the population consisted of men. Other research confirms the prevalence of males among those infected with leishmania [19,28], likely due to behavioral factors such as less strict adherence to preventive measures (i.e., repellent use) and more frequent exposure (i.e., outdoor activities for recreational and work purpose) [29]. It has also been hypothesized that smoking, which is more frequent among men, could be a risk factor for infection acquisition, as CO_2_ produced by cigarette combustion is an attractive substance for sandflies [30]. Moreover, biological sex has a relevant impact on physiology and immune response, consequently influencing the progression of disease [31]. Further research would be needed to clarify the role of biological sex in the clinical evolution of the disease.

Frequent outdoor activities and dog ownership were reported by 19.5% and 22.2% of the population, respectively, with no differences observed between the CL and VL populations. This result suggests that leishmaniasis should be considered within the diagnostic panel for individuals with a compatible clinical presentation, even in absence of evident exposure risk factors.

The median age was significantly higher (*p* < 0.005) in the VL group (63.8 [37–75.6]) compared to the CL group (38.3 [8–60.7]). Additionally, although not statistically significant, immunosuppression was more frequent in the VL group, where 31.4% patients were immunocompromised, compared to only 10.0% in the CL group. The age distribution of cases shows that the pediatric population <10 years is highly affected both by CL and VL. Our data reflect that older and immunocompromised populations, along with the pediatric population, are at higher risk for developing clinically evident forms of VL [32,33,34]. This is particularly important, as the elderly and the immunocompromised groups are growing in high-income countries, suggesting an increasing impact of the diseases on these vulnerable populations in the near future. HIV infection was reported in a minority of cases, confirming the previously described trend of VL gradually shifting from the HIV population to those with other immunosuppressive conditions, such as oncologic, autoimmune, hematological diseases and a history of solid organ transplant [35]. Compared to VL, the cutaneous disease shifts to younger age groups in the adult population, likely reflecting more frequent exposure due to different behavioral factors among the 30–40-year and 60–70-year groups.

The median time to diagnosis was 21 days for VL and 174 days for CL. Rapid diagnosis is crucial for early treatment and improving the outcome of VL, which can be fatal if left untreated. On the other hand, a significant delay in diagnosis of CL does not result in more severe outcomes due to the benign course of the disease at our latitudes. However, the long time needed for diagnosis highlights a lack of awareness about the presence of the disease in the study area, both among clinicians (that often do not consider CL among the differential diagnosis, leading to a diagnostic delay) and citizens (that are not aware of the presence of leishmaniasis in human at our latitude and do not know its clinical presentation).

Notably, time to diagnosis of VL was significantly longer in the immunocompromised population compared to the non-compromised. This is likely due to a wider spectrum of differential diagnosis in immunocompromised patients presenting with fever, splenomegaly and cytopenia, which can lead to considering leishmaniasis only after excluding other causes. Moreover, the clinical presentation in immunocompromised individuals can be atypical, making it easy to misdiagnose or mistake as a flare-up of the underlying disease. Therefore, awareness of leishmaniasis in the immunocompromised patient is of paramount importance, as this group is at higher risk of developing severe and recurrent forms of the disease [35].

This study showed high hospitalization rates (92.4%) and a median hospitalization duration of almost two weeks for the VL group. These results are consistent with previous data reporting leishmaniasis as the neglected tropical disease with the highest hospitalization rates in Italy, leading to substantial expenses for the national health system [36]. Complications were common, with 27.1% of subjects developing HLH, a life-threatening condition, followed by the need for blood component transfusion (19.5%) and bacterial superinfections (9.3%). Additionally, our results showed a 12.9% case fatality rate at 6 months for VL, and a 17.9% failure rate at 3 months of follow-up for CL. Of note, the deceased patients were significantly older compared to the median age of the study VL population, and 37.5% of them were immunocompromised. In this perspective, leishmaniasis should be addressed by future health policies as an emerging health concern that has a considerable impact on public health.

Of note, a relevant proportion of patients did not complete a 12-month follow-up, due both to personal reasons in some cases and shared clinical decision in others. Despite at least one year of follow-up being recommended by most experts both for CL and VL, this practice needs to be standardized among clinicians in the study area.

Reporting cases of human leishmaniasis to health authorities is mandatory in Italy. Moreover, a National Surveillance Plan for Leishmaniasis was issued in 2020, outlining the main surveillance and control strategies to be implemented at the human, vector and animal levels. The plan involves filling out a specific notification report that includes information on the clinical, epidemiological and diagnostic characteristics of each reported case [5]. The notification report was completed for 72.8% of observed cases in our study, although this result may be overestimated due to a selection bias; indeed, despite all efforts having been employed to detect all observed cases over the study period, some non-reported cases could have been missed. The issue of under-reporting in Italy has been highlighted in recent research, which reported a cumulative incidence of human leishmaniasis in Italy as 0.16 based solely on surveillance reports, but rising up to 0.70 when hospitalization reports are taken into account [19].

Similarly, despite not all cases of leishmaniasis, especially for CL, requiring hospitalization, surveillance data from the Regional Health Agency in Tuscany in 2023 indicated a higher number of cases detected through hospitalization reports, compared to those identified through mandatory notification reports [37]. Undoubtedly, the surveillance system needs strengthening in our country, as this represents the initial step toward obtaining reliable data on the epidemiological features and trends in the spread of the disease. This, in turn, would enable the implementation of appropriate and targeted control strategies.

## 5. Study Limitations

Among the limitations of this study, given the long incubation period of leishmaniasis and the fact that sandfly bites can pass unnoticed, information about the sites of infection may be imprecise. Moreover, due to the retrospective nature of this study, data about potential exposure sites and movements of each patient in the months preceding the disease onset could be incomplete. In this study, we established presumptive sites of infection based on suggestive clinical history (e.g., symptom onset following a stay in a specific vacation area). When no suggestive epidemiological elements were observed in the clinical history, we considered the domicile (i.e., the place where the patient spent most of their time) as the most probable site of infection.

Despite efforts made to reduce the underestimation of observed cases throughout the study period (see Section 2), many CL cases could have been missed for several reasons; for example, some cases might have gone undiagnosed and healed spontaneously. In addition, due to likely high under-reporting rates and the fact that many CL cases do not require hospitalization, some CL cases might have been managed exclusively by dermatologic outpatient clinics, thus being missed by our survey. The involvement and awareness of dermatologists will be essential for future epidemiological investigations of CL. Due to need of hospitalization and infectious disease referral, according to the methods used for data collection in this study (see Section 2), it is less probable that VL cases were missed in this study; however, implementation of data quality control mechanisms was not feasible for this study.

Due to lack of diagnostic standardization between centers and differences in access to healthcare in the region, only a portion of the patients completed a 12-month follow-up; therefore, data regarding follow-up of this study cohort are incomplete. Future interventions aimed at standardizing diagnosis and management of human leishmaniasis at regional and national level are needed in order to strengthen surveillance and control systems.

## 6. Conclusions

The geographical distribution of leishmaniasis is strongly influenced by environmental, climatic and socioeconomic factors, as well as changes in human habits and interactions with the environment. This study reveals an alarming increase in the disease’s prevalence in Tuscany, central Italy, from 0.22/100,000 in 2018 to 1.81/100,000 in 2023. Given its significant impact on the elderly and immunocompromised population, coupled with the observed high hospitalization (92.4% for VL, 40.0% for CL), mortality (12.9% at 6 months for VL) and treatment failure rates (at 3 months, 2.1% for VL, 17.9% for CL), this issue should garner increasing attention from public health policymakers. Enhancing awareness among healthcare workers and improving the surveillance system are crucial steps. Public awareness should also be increased, and education on personal protection measures (such as use of repellents) should be strengthened, especially among fragile groups such as children, the immunocompromised and the elderly. The epidemiological landscape of leishmaniasis in Central and Northern Italy warrants further investigation from a “One-Health” perspective. Geostatistical analyses are essential to understand potential relationships between the spatial–temporal spread of the diseases and climatic and environmental factors. Integrated research on the epidemiological characteristics of leishmaniasis across animal, vector and human levels would provide insights into the underlying mechanisms driving these epidemiological shifts. This knowledge is vital for guiding future efforts to implement effective control and surveillance strategies.

## Figures and Tables

**Figure 1 microorganisms-12-01963-f001:**
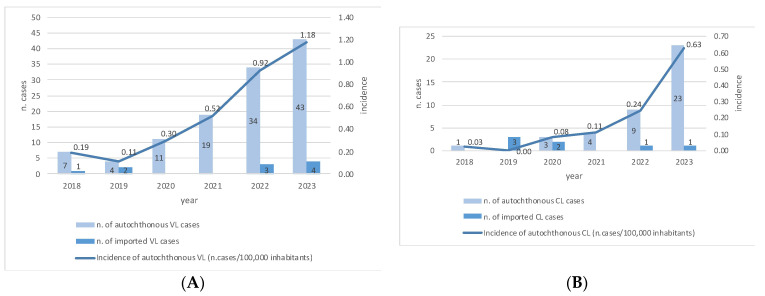
Absolute number of autochthonous and imported cases recorded each year and annual incidence of autochthonous VL (**A**) and CL (**B**) during the study period.

**Figure 2 microorganisms-12-01963-f002:**
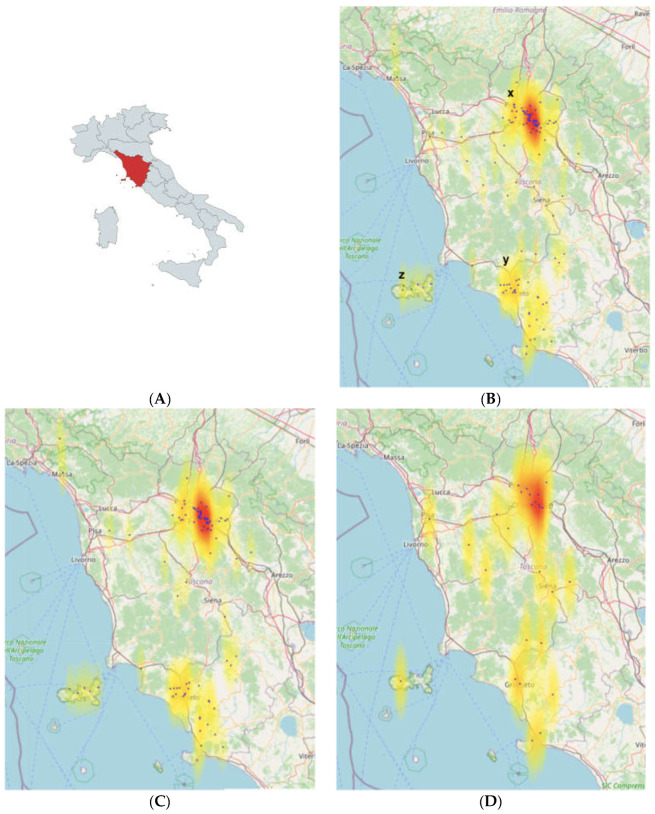
(**A**) Tuscany region (the study area) is represented in red in the Italian map; (**B**) overall distribution and Kernel density estimation of VL and CL cases; (**C**) distribution and Kernel density estimation of VL cases; (**D**) distribution and Kernel density estimation of CL cases.

**Figure 3 microorganisms-12-01963-f003:**
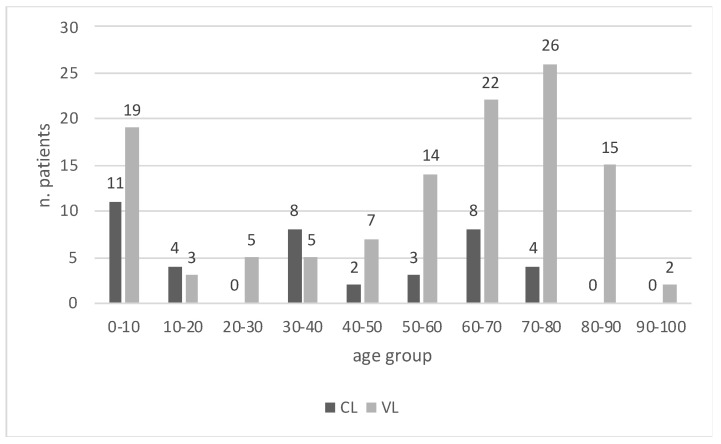
Case distribution by age group for autochthonous cases of CL and VL.

**Table 1 microorganisms-12-01963-t001:** Distribution of exposure risk factors and development of clinically evident infection risk factors among the subjects who acquired VL and CL in the study area; *p*-values were obtained through a logistic regression model.

	VL (tot. 118)	CL (tot. 40)	VL + CL (tot 158)	*p*-Value
Male sex	88 (74.6%)	26 (65.0%)	114 (72.2%)	0.694
Frequent outdoor activities	26 (22.0%)	3 (9.7%)	29 (19.5%)	0.240
Owning dogs	27 (22.9%)	6 (19.4%)	33 (22.2%)	0.619
Median age [IQR]	63.8 [37–75.6]	38.3 [8–60.7]	58.7 [25.3–73]	0.015
HIV infection	4 (3.4%)	1/35 (2.8%)	5/154 (3.3%)	0.547
Immunosuppression	37 (31.4%)	4 (10.0%)	41 (26.0%)	0.196

**Table 2 microorganisms-12-01963-t002:** Observed outcomes at 3, 6 and 12 months for autochthonous VL (left) and CL (right) cases.

VL	3 Months	6 Months	12 Months	CL	3 Months	6 Months	12 Months
Cure *	85 (90.4%)	66 (94.3%)	51 (99.9%)	Cure ^§^	23 (82.1%)	17 (100%)	11 (100%)
Failure ^#^	2 (2.1%)	2 (2.9%)	1 (1.9%)	Failure ^ç^	5 (17.9%)	0	0
Death	7 (7.5%)	2 (2.9%)	0	Death	0	0	0
Total	94	70	52	Total	28	17	11

* Cure for VL was defined as symptom regression and negativization of Leishmania PCR on peripheral blood after treatment initiation (evaluated at 3, 6 and 12 months). ^#^ Failure for VL was defined either as (1) persistence of symptoms (or worsening after an initial improvement) and detection of the parasite in blood or other biological samples through microscopy or PCR after treatment initiation or (2) reappearance of symptoms and detection of the parasite in blood or other biological samples through microscopy or PCR. ^§^ Cure for CL was defined as healing and re-epithelialization of skin lesions after treatment initiation. ^ç^ Failure for CL was defined either as (1) no improvement or worsening of skin lesions after treatment initiation or (2) reappearance of skin lesions after initial cure.

## Data Availability

The data presented in this study are available on request from the corresponding author due to privacy reasons.

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
