# Peer review of "The Uprise of Human Leishmaniasis in Tuscany, Central Italy: Clinical and Epidemiological Data from a Multicenter Study"

_microorganisms, 2024, doi:10.3390/microorganisms12101963_

Round 1
Reviewer 1 Report
Comments and Suggestions for Authors
Abstract
The term "socioeconomical factors" should be replaced with "socioeconomic factors," which is more commonly used and precise. It is worth strengthening the conclusions with more specific action proposals that should be taken. There are unnecessary elements in the abstract, such as citation references (e.g., Citation, Microorganisms 2024). These should be removed to maintain the clarity and transparency of the content.
Introduction
The text suggests that the number of leishmaniasis cases is likely underestimated due to a lack of comprehensive reporting (line 72: "likely high rate of under-reporting"). More details should be provided on why reporting is incomplete and what factors contribute to underreporting. The text mentions that dogs are the main reservoirs of leishmaniasis in the region (line 86: "Historically, the main reservoir for the infection has been the dog..."). However, later it states the potential role of wild mammals as additional reservoirs (line 90: "highlighted the potential role of sylvatic mammals as additional animal reservoirs..."). This issue should be expanded upon, and the specific species of wild mammals that may play a role in the transmission of leishmaniasis and how this affects the disease dynamics should be discussed. The text indicates that VL has a high mortality rate if left untreated (line 76: "fatality rates, which can reach 95% if untreated"). More details should be included regarding the availability and effectiveness of treatment in regions where leishmaniasis is endemic, as well as the main challenges in controlling the disease. The mention of leishmaniasis distribution across different regions of Italy is relatively general (line 84: "Southern regions, the major islands, the Tyrrhenian coast and, to a lesser extent, the Adriatic coast"). More specific data should be provided about the regions and identify the specific areas in the northern part of the country most affected by the spread of the disease. The text notes an increase in VL cases in Tuscany in 2021 and 2022 (line 95: "Annual VL cases ranged from 2 to 7 until 2020 but increased to 14 in 2021 and 26 in 2022"). Potential reasons for this increase should be discussed. Understanding the reasons for the rise may help explain the disease dynamics.
Materials and Methods
It is worth noting whether the REDCap tool (line 122) was properly configured for multi-center data processing and whether data validation was conducted. REDCap is a powerful tool, but some details regarding its use could enhance the credibility of the study results. Although the authors mention a strategy to reduce underreporting by collecting data from laboratories and reports (line 115), it would be beneficial to discuss how effective this method was. Are there known cases that may have been missed despite these efforts? Is there a data quality control mechanism or case verification process in place? The authors mention the diagnosis of HLH based on criteria (line 131), but it is unclear how many such cases were recorded and how they were related to leishmaniasis. The number of HLH cases in the studied group should be mentioned. The section lacks a discussion of the potential limitations of the study. Possible sources of error should be identified, such as the lack of diagnostic standardization between centers, differences in access to healthcare in the region, or the lack of full data consistency from different sources (e.g., discrepancies between reported cases and actual hospitalizations). The description of the statistical methods (lines 134-140) is clear, but it would be useful to add details regarding which specific variables were analyzed for associations. For example, were relationships between age, gender, and type of leishmaniasis analyzed? Was a multivariate analysis considered to account for potential confounding variables?
Results
The description of the results sometimes lacks clear categorization of information. For example, when describing the number of VL and CL cases from 2018–2023, a more organized presentation of numerical data could improve readability. Only part of the patients completed the full follow-up period (12 months). This significantly reduces the sample size and may lead to incomplete conclusions. Although the number of deaths is provided (9 deaths), the details about their causes are only partially described. Only 7 deaths had attributed causes, presenting an incomplete picture. The criteria for assessing the success or failure of VL and CL therapy are clearly defined, but there is no fully consistent way of comparing these outcomes between different patient groups. For example, introducing a standard success rate indicator that takes into account other factors such as age and immunosuppression status would help better understand the dynamics of treatment outcomes in different groups.
Discussion
One part of the text mentions that leishmaniasis is on the rise in Tuscany, contrary to national reports indicating a decline in cases. It is important to clearly indicate that these differences may be due to local ecological and climatic factors. It should also be emphasized that national-level data showing a decline in cases may mask regional differences, which should be discussed more broadly. The text mentions the role of climate change in expanding sandfly habitats, consistent with the literature. However, this section lacks a detailed analysis of the specific climatic changes that may have affected Tuscany. Specific meteorological data should be added to support this hypothesis for the studied location. The chapter indicates that 74.7% of cases are of the visceral form (VL), consistent with the literature. However, there is a lack of deeper discussion on why the cutaneous form (CL) may have been underestimated. The text suggests that urbanization and land use changes may be factors influencing epidemiological changes. However, there is a lack of specific data on this topic concerning Tuscany. It would be helpful to provide specific examples of land use changes that may affect disease transmission in this area. The median time to diagnosis for VL of 21 days and 174 days for CL is a significant issue, highlighting the need for greater awareness among physicians. However, there is no discussion of why the time to diagnosis for CL is so long.
Conclusions
The chapter contains several general statements, such as "alarming increase," "high hospitalization, mortality, and treatment failure rates," but lacks specific data to support these statements. Providing percentages, numbers, or statistics could better justify these claims and underscore the seriousness of the problem.
Reviewer 2 Report
Comments and Suggestions for Authors
I suggest that in the title of the work you mention that it is a retrospective study.
The methodology and results are clearly presented and follow a logical sequence that facilitates understanding of the research.
It is a very complete research work, especially in the discussion, they analyzed the results in a very thorough manner, comparing them with those previously reported, and they gave a possible explanation for the result found.
The conclusion seems to me to be supported by the results found in this research.
It seems to me that the number of co-authors is excessive for this type of research studies.
Round 2
Reviewer 1 Report
Comments and Suggestions for Authors
Accept in present form